# Application of Silicon, Zinc, and Zeolite Nanoparticles—A Tool to Enhance Drought Stress Tolerance in Coriander Plants for Better Growth Performance and Productivity

**DOI:** 10.3390/plants12152838

**Published:** 2023-07-31

**Authors:** Abdel Wahab M. Mahmoud, Hassan M. Rashad, Sanaa E. A. Esmail, Hameed Alsamadany, Emad A. Abdeldaym

**Affiliations:** 1Plant Physiology Division, Department of Agricultural Botany, Faculty of Agriculture, Cairo University, Giza 12613, Egypt; mohamed.mahmoud@agr.cu.edu.eg; 2Department of Biological Sciences, Faculty of Science, King Abdulaziz University, Jeddah 21589, Saudi Arabia; hali@kau.edu.sa (H.M.R.); halsamadani@kau.edu.sa (H.A.); 3Department of Ornamental Horticulture, Faculty of Agriculture, Cairo University, Giza 12613, Egypt; sanaa.ahmed@agr.cu.edu.eg; 4Department of Vegetable, Faculty of Agriculture, Cairo University, Giza 12613, Egypt

**Keywords:** nano-elements, water shortage, *Coriandrum sativum*, plant performance, physiological responses, biochemical properties, oil composition

## Abstract

Drought stress in arid regions is a serious factor affecting yield quantity and quality of economic crops. Under drought conditions, the application of nano-elements and nano-agents of water retention improved the water use efficiency, growth performance, and yield quantity of drought-stressed plants. For this objective, two field experiments were performed and organized as randomized complete block designs with six replications. The treatments included kaolin (5 t. ha^−1^) bentonite (12.5 t. ha^−1^), perlite (1.25 t.ha^−1^), N-zeolite (1.3 L.ha^−1^), N-silicon (2.5 L.ha^−1^), and N-zinc (2.5 L.ha^−1^). The current study showed that the application of silicon, zinc, and zeolite nanoparticles only positively influenced the morphological, physiological, and biochemical properties of the drought-stressed coriander plant. Exogenous application of N-silicon, N-zinc, and N-zeolite recorded the higher growth parameters of drought-stressed plants; namely, plant fresh weight, plant dry weight, leaf area, and root length than all the other treatments in both seasons. The improvement ratio, on average for both seasons, reached 17.93, 17.93, and 18.85% for plant fresh weight, 73.46, 73.46, and 75.81% for plant dry weight, 3.65, 3.65, and 3.87% for leaf area, and 17.46, 17.46, and 17.16% for root length of drought-stressed plants treated with N-silicon, N-zinc, and N-zeolite, respectively. For physiological responses, the application of N-zeolite, N-silicon, and N-zinc significantly increased leaf chlorophyll content, photosynthetic rate, water use efficiency, chlorophyll fluorescence, and photosystem II efficiency compared with the control in both seasons, respectively. Similar results were observed in antioxidant compounds, nutrient accumulation, and phytohormones. In contrast, those treatments markedly reduced the value of transpiration rate, nonphotochemical quenching, MDA, ABA, and CAT compared to control plants. Regarding the seed and oil yield, higher seed and oil yields were recorded in drought-stressed plants treated with N-zeolite followed by N-silicon and N-zinc than all the other treatments. Application of N-zeolite, N-silicon and N-zinc could be a promising approach to improve plant growth and productivity as well as to alleviate the adverse impacts of drought stress on coriander plants in arid and semi-arid areas.

## 1. Introduction

The global demand for food is rising annually due to the rapid growth of the world population. Water shortage is considered one of most the serious factors affecting food production and human health, particularly in arid and semi-arid areas [1,2]. Water shortage is also increasing in various parts of the world, which limits agricultural production [3]. Decreasing water resources globally is associated with climate change, mishandling of water resources, and declining rainfall; these factors have unfavorable impacts on plant growth and production [4,5]. Water shortage, known also as a drought stress, for a short time can severely impact growth and development as well as reduce yield quantity and quality [6,7]. Several reports stated that water stress can cause dehydration of plant cells, reduction in nutrient absorption, disruption of plant hormone production, damage cell membrane permeability of plants, as well as decrease photosynthesis rate and carbon dioxide assimilation due to stomata closure [8,9]. Khan et al. [10] confirmed that water stress is accountable for the yield reduction in food crops along with imbalanced fertilization, weeds, soil salinity, and nutrient deficiency.

Plants have some approaches to alleviate the negative impacts of drought stress by inducing physiological, biochemical, and morphological alterations [7,9]. Among these changes is the accumulation of organic osmolytes, which is of substantial prominence as it conserves osmotic pressure in the plant cell and prevents water loss under drought stress conditions [9,11]. This accumulation contributes to stabilizing the structure of different biomolecules. Improving water use efficiency (WUE) and organic osmolytes accumulation in crop plants under water stress conditions is key to enhance plant growth and production [7,11].

In this regard, several agronomic strategies have been suggested to improve WUE and yield production of water stressed plants, for example, application of nanofertilizers [12], utilization of microorganisms [7], soil addition of organic fertilizers and biochar [13,14], use of tolerant rootstocks and cultivars [8,9], anti-transparent substances application [15], implementation of material with high water retention [16,17], and application of efficient irrigation methods, particularly in soils characterized by low water hold capacity [12].

Application of certain nutrients, either foliar or soil addition, improves plant tolerance against abiotic stresses such as water stress, salinity, and other biotic stresses [18,19,20]. Application of zinc (Zn) and silicon (Si) decreases water stress [18,20,21,22]. Under water stress, both elements play vital roles in the improvement of organic osmolyte accumulation, water balance in plants, rate of photosynthesis, stomata movements, production of endogenous phytohormones, nutrient uptake, accumulation of primary and secondary metabolites which eventually promote the plant growth of plants, biomass accumulation, as well as yield quantity and quality [20,23]. At the same time, many scientific reports confirmed that these elements (Si and Zn) significantly reduced the activity level of reactive oxygen species (ROS) and improved the total antioxidant activity and activity levels of antioxidant enzymes under environmental stresses such as drought stress [18,19,20,21,22,23].

Nanotechnology has gained great attention in recent decades due to its increased number of applications. Nanoparticles are characterized by solubility, surface area, and reactivity compared to bulk material [24,25]. Furthermore, nanoparticles have acquired a promising status to improve the adverse impacts of environmental stressors, such as drought stress, salinity, and nutrient deficiency, to attain the objective of sustainable agriculture [26]. Due to their influence on environmental stress tolerance and the nutritional value of crops, studies associated with nanoparticle implementation are increasing. Many nanoparticles have been examined for their protective impact against environmental stresses and other biotic stresses [26,27,28].

Furthermore, numerous studies have been carried out on different plants’ adaptations to drought stress; however, little information is available about drought stress-reducing factors.

Several scientists confirmed that macro and microelement application in nanoform (nano-elements), such as potassium, boron, zinc, and silicon, significantly improved crop yield and plant tolerance against abiotic stresses [17,18,19,20]. The tiny size of nano-elements is responsible for higher uptake by plants than conventional ones under stress conditions [26], which consequently improve the physiological and biochemical processes of the stressed plants [18,19].

Furthermore, application of water retention agents, (i.e., zeolite, bentonite, and perlite), either bulk or nanoform, to soil in order to enhance soil physical properties is considered an important method to reduce drought stress [18,28,29,30]. In previous studies, findings confirmed that soil application of water retention agents, such as zeolite, improved moisture and water storage of soil as well as enhanced water use efficiency and yield production for different plants [31,32]. Hence, application of this material is considered to be an applicable and low-cost method for plants to alleviate environmental stresses such as drought [32]. Some positive effects of zeolite on agro-physiological and biochemical measurements of plants, i.e., plant fresh weight, plant dry weight, nutrient content, photosynthesis rate, leaf chlorophyll content, relative water content, plant hormone production, antioxidant compounds, yield quantity and quality, under environmental stress conditions [30,31,32,33]. Therefore, the objective of this study is to use kaolin, bentonite, perlite, -zeolite nanoparticles (N-zeolite), silicon nanoparticles (N-silicon), and zinc nanoparticles (N-zinc) to reduce the impacts of drought stress on coriander plants. In addition, it also assesses the impacts of these treatments on morphological, physiological, and biochemical traits, as well as the quantity of fruit, seeds, and oil in water-stressed plants.

## 2. Results

### 2.1. Vegetative Growth Parameters of Coriander Plants

Data in Figure 1A–F show that the application of bentonite, perlite, kaolinite, N-silicon, N-zinc, and N-zeolite significantly influenced growth measurements of coriander plants under drought stress conditions. The studied measurements were plant height, number of umbels, shoot fresh weight, shoot dry weight, leaf area, and root length. In both seasons, plant height and number of umbels per plant were greater in coriander plants treated with N-zeolite treatment than all other treatments, while non-significant differences were in plant height in the remaining treatments and untreated plants (Figure 1A).

Likewise, compared with the untreated plants, the maximum values of root length, fresh weight, and dry weight were observed in coriander plants treated with N-zeolite followed by N-silicon and N-zinc, while the minimum values were in plants treated with perlite, with non-significant differences between untreated plants and plants treated with bentonite and kaolinite (Figure 1B–F) under water stress conditions.

### 2.2. Leaf Chlorophyll Content, Photosynthetic Machinery, Photosystem II Efficiency, Chlorophyll Fluorescence, and Total Carbohydrate Content

Table 1 shows the effect of bentonite, perlite, kaolinite, N-silicon, N-zinc, and N-zeolite on chlorophyll content, photosynthetic machinery, photosystem II efficiency, and chlorophyll fluorescence under water stress conditions. However, the application of N-zeolite-n, N-silicon, and N-zinc significantly improved total chlorophyll content, photosynthetic rate, intercellular CO_2_ concentration, and water use efficiency and reduced transpiration rate compared to untreated plants (control). The plants treated with kaolinite and bentonite did not show any significant changes in photosynthetic machinery.

The maximum values of the chlorophyll content, photosynthetic rate, intercellular CO_2_ concentration, and water use were recorded in plants treated with N-zeolite application followed by N-silicon and N-zinc, while the minimum values were noted in plants treated with perlite material. In contrast, the highest transpiration rate was observed in untreated plants and plants treated with perlite material compared with all other treatments.

Similar results were observed in treated plants in photosystem II efficiency (ϕPSII), chlorophyll fluorescence (Fv/Fm), nonphotochemical quenching (NPQ), and total carbohydrates of coriander plants grown under drought stress conditions (Figure 2). Compared with control plants, application of N-zeolite-n, N-silicon, and N-zinc enhanced the leaf Fv/Fm by 17.64, 10.49, and 6. 97% in addition to raising the ϕPSII by 12.01, 2.34, and 7.02%, respectively, in over two seasons (Figure 2A,B). Furthermore, the previous treatments markedly increased carbohydrate content by 25.80, 19.60, and 11.70%, respectively (Figure 2D). Otherwise, bentonite and kaolinite applications failed to modify Fv/Fm, ϕPSII, and carbohydrate content. The lowest values of Fv/Fm, ϕPSII, and carbohydrate content were achieved in plants treated with perlite materials compared to all other treatments (Figure 2A,B,D). On the other hand, the highest values of NPQ were achieved in plants supplied with perlite, bentonite, and kaolinite while the lowest value was observed in plants treated with N-zeolite (Figure 2C).

### 2.3. Antioxidant Compounds Content and Proline

Data in Figure 3 illustrated that the concentration of proline and antioxidant compounds (i.e., carotenoids, total phenols, flavonoids, ascorbic acid, and thiamine) were progressively impacted by drought stress (Figure 3). All coriander plants supplied with N-zeolite, N-silicon, and N-zinc markedly enhanced the concentration of carotenoids, total phenols, flavonoids, ascorbic acid, thiamine, and proline in the leaf tissues (Figure 3A–F). Compared with the control plants, maximum values of the aforementioned parameters were observed in drought-stressed coriander plants while the lowest values were recorded in plants treated with perlite materials in both seasons.

### 2.4. Phytohormones, Malondialdehyde, and Catalase

Considering the corresponding control, the concentrations of leaf IAA and GA3 significantly increased in response to N-zeolite, N-silicon, and N-zinc application while these concentrations reduced due to perlite application (Table 2). However, N-zeolite, N-silicon, and N-zinc applications significantly augmented the content of IAA in leaves by 28.48, 24.44, and 19.24% in the first season and 26.12, 15.3, and 13.97% in the second season, respectively, compared with control plants. Such a result was noted in GA3 concentration in plants treated with N-zeolite, N-silicon, and N-zinc. The increment ratio in GA3 concentration reached 21.44, 17.08, and 11.50% in the first season and 17.35, 10.89, and 6.60% in the second season, respectively, against untreated plants (control). Meanwhile, these treatments significantly reduced the leaf ABA and MDA content in addition to the activity level of CAT enzymes in comparison to control plants. On the other hand, the plant treated with the perlite treatment showed a significant reduction in leaf IAA and GA3 content and elevation in activity level of CAT enzyme and contents of leaf ABA and MDA content compared to the control treatment.

### 2.5. Macro and Micronutrient Content

Results in Table 3 pointed out the impact of different treatment applications on macro and microelement accumulation in the tissues of drought-stressed plants. The coriander plants supplied with N-zeolite, N-silicon, and N-Zinc suggestively improved the accumulation of endogenous microelements (N, P, K, Mg, and Ca) and micronutrients (Fe, Mn, and Zn) in their tissues compared with untreated plants (control). Under drought stress, the highest concentrations of N, P, K, Mg, Ca, Fe, and Zn in plant tissue occurred when N-zeolite was applied, while the lowest values were observed in plants treated with perlite. Otherwise, both bentonite and kaolinite applications failed to change macro and micronutrient accumulation in plant tissues.

### 2.6. Essential Oil, Fruit Yield, and Its Components in Coriander Plants

Data displayed in Table 4 indicated that the greatest values of fruit yield, seeds weight/1000 plant, and essential oil were recorded in drought-stressed plants treated with N-zeolite followed by N-zinc and N-silicon, and the lowest values were observed in plants treated with perlite compared with untreated plants, without significant differences between bentonite, kaolinite, and control plants. Compared with untreated control plants, the improvement ratios in fruit yield per plant, seeds weight/1000, fruit yield per feddan, essential oil per plant, and essential oil per feddan reached 17.17, 11.33, 13.66, 24.24, and 25.30% for N-zeolite treatment; 8.71, 8.22, 10.38, 16.67, and 17.33% for N-zinc treatment; and 8.23, 5.85, 7.84,13.79, and 4.91%, respectively, in the first season. Similar trends of enhancement ratios were observed in the second year for aforementioned yield parameters, whereas they reached 17.41, 14.70, 13.33, 18.75, and 25.01% for N-zeolite treatment; 9.71, 11.98, 10.52, 13.33, and 16.49% for N-zinc treatment; and 9.23, 9.59, 7.60,11.86, and 3.13%, respectively.

On the contrary, compared with control plants, perlite treatment caused a reduction in fruit yield, seeds weight/1000 plant, and essential oil of coriander plants by 5.83, 15.93, 4.41, 19.05, and −13.61% in the first year and 7.53, 8.68, 7.11, 30, and 11.51%, respectively, in the second year (Table 4).

### 2.7. Composition of Essential Oil

The GC/MS analysis and heatmap correlation (Table 5 and Figure 4) revealed 11 fatty acid compounds in the essential oil of coriander (Table 5). The main essential oil compounds were linalool, β-Pinene, limonene, p-cymene, α-pinne, and camphor in both seasons. Linalool, as the main fatty acid composite, was affected by the application of N-silicon, N-zinc, and N-zeolite. The highest values of the linalool compound were recorded in drought-stressed plants treated with N-zeolite followed by N-Zn compared with control plants. Similar results were observed in β-Pinene, limonene, p-cymene, α-pinne, and camphor compounds in both seasons. The maximum concentrations of the aforementioned fatty acid compounds were observed in drought-stressed plants supplied with N-zeolite followed by N-Zn and N-Si, while the minimum values of the previous fatty acid compounds were noted in the untreated plants. On the contrary, the greatest concentration of nerol, camphene, sabinene, myrcene, and borneol was recorded in untreated plants than all the other plants.

### 2.8. Correlation Study

Pearson’s correlation analysis and heatmap correlation show the changes in physicochemical assets of coriander plants grown under drought stress conditions (Figure 5 and Figure 6). The heatmap based on the 27 parameters clearly classified them into two groups (A and B), while the bentonite, perlite, and kaolinite treatments were inserted together under group A. These treatments were closest in efficiency to the control treatment (Figure 5). Meanwhile, the second group (B) included the N-silicon, N-zinc, and N-zeolite treatments. The red color indicates a positive effect, and the blue color shows a negative effect. The heatmap correlation also indicates that the N-zeolite treatment has a more positive effect on studied parameters except for CAT, MDA, and transpiration rate; in contrast, perlite treatment has more negative effects.

Similarly, Pearson’s correlation analysis was used to detect the positive and negative correlations between the studied measurements. A positive correlation (blue color) and negative relationships (red color) are shown in Figure 6. Pearson’s correlation analysis displayed that fruit yield positively correlated with LA, Shoot FW, Shoot DW, Chl, Wt. 1000 seeds, EO, Pn, Int.CO_2_.Conc, WUE, SC, Fv/Fm, PSII, and Tot. Carboh content. A similar relationship also was observed between fruit yield and antioxidant compounds (Caro, TPC, TFC, and ASA). Furthermore, fruit yield is positively associated with Pro, IAA, and GA3. On the contrary, fruit yield is negatively linked to Tr, ABA, and CAT.

## 3. Discussion

Coriander plants are susceptible to drought stress, and irrigation is considered the main source of water in tropical and subtropical regions affecting growth performance, seed quality, and oil yield [32]. The result of the current study showed that the plant growth parameters (plant height, fresh weight, dry weight, leaf area, and root length) markedly reduced in untreated plants under drought stress conditions (Figure 1). Recent reports have presented the adverse effects of drought on different economic crops such as cucumber [7,8], tomato [9,12], cauliflower [9], maize [14], and wheat [10,13,15]. These reductions in plant growth performances under drought stress conditions might be associated with declined cell division and elongation due to damage of turgidity, decreased photosynthesis, and reduced energy input [34,35]. Conversely, application of N-silicon, N-zinc, and N-zeolite improved the studied growth traits, namely plant height, fresh weight, dry weight, leaf area, and root length, particularly under drought stress (Figure 1). This could be linked to the enhancement of water use efficiency, nutrient accumulation, photosynthesis rate, and phytohormones (IAA and GA3) as exhibited in this study ((Figure 5), which provides a better condition for plant growth and development [22,27]. Furthermore, Pearson’s correlation analysis confirmed that fresh weight, dry weight, and leaf area connected positively with chlorophyll content, photosynthesis rate, stomatal conductance, water use efficiency, IAA, and GA3 (Figure 6).

The chlorophyll content and chlorophyll fluorescence are vital indicators for the photosynthesis activity of a plant [36,37]. Chlorophyll fluorescence analysis is considered one of the most important techniques used to evaluate the effect of biotic and abiotic stresses on plants which is one of the results of light absorption by plants. The PSII is a sensitive factor of the photosynthesis system concerning water deficit stress [36].

As shown in Table 2, leaf chlorophyll content, chlorophyll fluorescence (Fv/Fm), photosystem II efficiency (ϕPSII), and other photosynthetic measurements, including photosynthetic rate, intercellular CO_2_ concentration, and water use efficiency of untreated plants (Table 4 and Figure 3) significantly reduced under drought stress conditions. The reduction in chlorophyll content in untreated drought-stressed plants could be associated with increasing the activity level of the chlorophyllase enzyme and/or the inhibition of photosynthetic pigment formation [38]; this consequently reduced leaf photosynthesis rate, stomatal conductance (SC), water use efficiency, and photosystem II efficiency. In addition, Pearson’s correlation analysis shows a positive relationship between chlorophyll content and leaf photosynthesis rate (Figure 6). Numerous studies have indicated that a reduction in photosynthesis rates is not only predominantly due to the decline in leaf chlorophyll content but is also associated with decreasing stomatal conductance of leaves, which decreases the supply of carbon dioxide into the intercellular spaces [39,40]. Contrariwise, the application of silicon, zinc, and zeolite evidently reduced leaf chlorophyll degradation and increased SC and Fv/Fm and photosynthetic apparatus of drought-stressed plants [41,42,43,44,45]. These findings could be related to the ability of the previous nano-treatments to alleviate the negative impacts of drought by enhancing hydraulic conductivity, conserving higher transpiratory and photosynthesis rates, photosynthetic pigment concentrations, and reducing oxidative damages [46,47]. Furthermore, the improvement in total carbohydrate content in drought-stressed plants supplied with N-silicon, N-zinc, and N-zeolite could be associated with elevating the photosynthesis rate and CO_2_ assimilation, as shown in Figure 2D [20,41,48,49,50]. Pearson’s correlation analysis showed that total carbohydrate content correlated positively with photosynthesis rate, intercellular CO_2_ concentration, and water use efficiency (Figure 6).

Water-deficient stress markedly influenced phytohormone concentration, especially IAA, GA3, and ABA (Table 2). Our findings showed that N-silicon, N-zinc, and N-zeolite caused an improvement in the levels of IAA and GA3 and a reduction in the ABA level in the leaves of coriander plant under drought stress conditions (Table 2 and Figure 5). These findings were in harmony with results reported by Othman et al. [50] and Umair Hassan et al. [51], who found that application of silicon, zinc, and zeolite nanoparticles upregulated the concentration of IAA and GA3 and downregulated ABA concentration in leaves of different plants exposed to drought stress. This result could be attributed to the improved water and nutrient uptake, especially Zn. Improving the accumulation of Zn in plant tissues (Table 6) plays an important role in biosynthesizing tryptophan. This amino acid is considered a fundamental compound for IAA formation within the plant [51]. Regarding leaf GA3 content, the improvement of GA3 is mostly linked to IAA upregulated in plants [52]. Furthermore, the current study has proven that there is a strong correlation between leaf IAA content and GA3 content (Figure 6). Some investigators also confirmed that IAA activation promoted GA3 biosynthesis [51].

The results of the present study displayed that water shortage clearly upgraded the malondialdehyde level (MDA) in the leaves of untreated plants compared with plants treated with N-silicon, N-zinc, and N-zeolite (Table 2). MDA is considered one of the final products of polyunsaturated fatty acid decomposition in plant cell membranes [53]; therefore, oxidative injury of lipids in cell membranes of plants is determined by high MDA levels in plant tissues [53]. Subsequently, augmented lipid peroxidation and hydrogen peroxide levels increase oxidative pressure due to increasing reactive oxygen species (ROS) and interruption of the enzymatic defense in plants grown under water limitation conditions [53,54]. Our results are in harmony with the findings of some researchers who stated that malondialdehyde significantly increased under drought stress in various plants [55]. The current study shows that the application of N-silicon, N-zinc, and N-zeolite on coriander pants alleviated the harmful impacts of drought stress by removing damage caused by oxidative stress and protecting the plant cell through their capability to reduce water and nutrient loss from plants as well as improve the water-hold capacity of soil through zeolite nanoparticle (N-zeolite) application [41,42,43,45,50].

Under drought stress conditions, plants produce and accumulate efficient antioxidant compounds to reduce ROS activity [56]. Those compounds, namely carotenoids, total phenols, flavonoids, ascorbic acid, and thiamine, also play an important role as non-enzymatic free radical scavengers by deactivating singlet oxygen or/and reducing metal ions to safeguard the plant cells exposed to oxidative damages [7,8,9,11,18,19,50,51,52]. The obtained results show an increase in the level of antioxidant compounds (carotenoids, total phenols, flavonoids, ascorbic acid, and thiamine) in leaves of coriander plants treated with N-silicon, N-zinc, and N-zeolite than untreated plants in order to hamper their harmful impacts (Figure 3). In agreement, Othman et al. [50] and Maghsoudi et al. [37] declared that the application of N-silicon, N-zinc, and N-zeolite increase non-enzymatic antioxidants and reduce MDA levels. Numerous scientists revealed that elevated antioxidant enzymes (SOD, CAT, and POD) are associated with increased levels of MDA and H_2_O_2_ in drought-stressed plants [7,8,11,50]. A correlation study confirmed that MDA correlated negatively with antioxidant compounds [carotenoids (Caro), total phenols content (TPC), flavonoids (TFC), ascorbic acid (AsA), and thiamine (TAM)], as presented in Figure 6.

Under drought stress conditions, plants elevate suitable solutes, such as proline, in their cells for assisting water absorption, reducing cell destruction, and improving the osmotic potential of plant cells [57]. The accumulation of proline is a general indicator of drought stress tolerance and permits osmotic modification that results in cell dehydration prevention and water retention [58]. In this study, drought-stressed plants treated with N-silicon, N-zinc, and N-zeolite showed higher proline concentrations than untreated plants. Proline accumulation under environmental stress conditions in different crops has been associated with tolerance to stressors, and the proline level is greater in tolerant plants than in susceptible plants [59].

On the contrary, applying the silicon, zinc, and zeolite nanoparticles led to a reduction in the activity level of CAT and MDA contents. These results can be explained by the involvement of silicon, zinc, and zeolite nanoparticles to reduce water and nutrient losses, hence a decline in the oxidative damage of plant cells [50,51]. Comparable results were observed in salt-stressed potato plants exposed to silicon, zinc, and zeolite nanoparticles with reduced ABA and antioxidant enzymes (CAT and POD) [18,19].

Likewise, water limitations considerably reduced the accumulation of nutrients (N, P, K, Ca, Mg, Zn, and Fe) in leaf tissues of control plants compared with treated plants with N-silicon, N-zinc, and N-zeolite, as shown in Table 3. The highest accumulation of endogenous nutrients was recorded in the leaf tissue of plants supplied with N-zeolite followed by N-silicon and N-zinc (Table 6). This enhancement in the aforementioned nutrients in leaf tissues might be due to the increasing root length and hydraulic conductivity of drought-stressed plants (Figure 1F) supplied with N-zeolite, N-silicon, and N-zinc, and thus, upgraded the nutrient uptake by the plant [53,54].

In terms of coriander productivity, the attained results indicated that the application of N-silicon, N-zinc, and N-zeolite increased seed and oil yield of coriander under drought stress conditions, particularly the weight of 1000 seeds as well as quantity and quality of essential oil, as shown in Table 4, Table 5, and Figure 4. These results were in line with the findings reported by Ghamarnia1 and Daichin [33] and Afshari et al. [60] who revealed that the application of N-silicon, N-zinc, and N-zeolite improved the seed and oil yield of coriander plants. The improvement in the quantity of seed and essential oil yield of drought-stressed plants treated with nano-treatments (silicon, zinc, and zeolite) could be related to the enhancement of nutrient uptake, photosynthesis rate, PSII, Fv/Fm, and water use efficiency, which consequently increased fruit yield and weight of 1000 seeds as well as improved the accumulation of secondary metabolites and essential oil in seeds [61]. In contrast, the EOs yield of untreated drought-stressed plants was significantly reduced due to a high decrement in seed yield. A correlation study showed that EO yield correlated positively with weight per 100 seeds, fruit yield, photosynthesis rate, PSII, Fv/Fm, and WUE (Figure 6).

Furthermore, the composition of essential oil in treated drought-stressed plants differed from untreated plants. For instance, the application of N-silicon, N-zinc, and N-zeolite elevated the percentage of linalool, β-Pinene, limonene, p-cymene, α-pinne, and camphor and reduced nerol, camphene, sabinene, myrcene, and borneol percentage compared with the control (Table 5 and Figure 4). A change in the amount and composition of essential oil is influenced by genetic and environmental stressors, particularly high temperature, salinity, and drought stress. The severity of environmental stressors can also impact on the main compounds of essential oils [61,62]. Meanwhile, some scientific investigators reported that the coriander plants exposed to moderate drought stress reached optimum quantity and quality of essential oil more than plants grown in severe drought stress [60]. In this study, the application of zielote, SA, and Zn may have a stimulatory impact on the expression of the regulatory enzyme (limonene synthase) of the EO biosynthetic pathway, which possibly increases the EOs yield and the concentration of major compounds in the seed of coriander plants [60,61,62].

In general, the application of zeolite, silicon, and zinc nanoparticles in drought-stressed coriander plants stimulates plant tolerance against drought stress by alleviating the adverse impacts of drought by enhancing water use efficiency and the photosynthetic rate, decreasing the oxidative damages, regulating phytohormones. In addition, these nano-treatments also improve nutrient accumulation and the activity of enzymatic and non-enzymatic antioxidants, which eventually augment the plant growth performance, quantity of seeds, and EOs for coriander plants.

## 4. Materials and Methods

### 4.1. Experimental Site, Fertilizers Application, and Plant Materials

Two field experiments were conducted in a private farm (sandy soil) at Ismailia Governorate, 150 km from the northeastern part of Cairo city, Egypt. (Latitude: 30°36′15.37″ N, and Longitude: 32°16′20.10″ E) in 2021 and 2022. Before seed sowing, physicochemical analyses of the soil (Table 6) and chemical analysis of applied compost (Table 7) were performed at the Soil, Water, and Environment Research Institute (SWERI), Agriculture Research Center (A.R.C) according to [63,64,65]. Compost, at the rate of 2.3 t. ha^−1^, was incorporated into the soil 15 days before planting. The total annual precipitation was 2.52 mm in the first season and 2.80 mm in the second season.

Seeds of *Coriandrum sativum* L. at the rate of (25 kg per fadden) were obtained from the nursery of the Faculty of Pharmacy, Cairo University, and sown on the 10th February in both years (2021 and 2022). The area of each plot was 6 m^2^. The space between 2 lines and between plants on the same line was 30 cm and 20 cm, respectively. The space between plots was 100 cm.

Plants were harvested 30 days from seed sowing. Recommended chemical fertilizers according to the Ministry of Agriculture and Land Reclamation (MALR, Egypt) were added at the rate of 100 kg ammonium sulphate (20.5% N) 3 days before planting and repeated after every cut. Then, 3 kg potassium sulphate (48% K) and 25 kg mono-superphosphate (15.5% P) were added only once during soil preparation. Seeds of the coriander plant were collected from each treatment separately at the end of June for years 2021 and 2022.

### 4.2. Experimental Design and Treatments Applied

The experiment was organized as a randomized complete block design (RCBD) with 7 treatments, which were as follows (1) untreated plants (control), (2) bentonite, (3) perlite, (4) kaolinite, (5) N-silicon, (6) N-zinc, and N-zeolite in 2021 and 2022. Each treatment was replicated 4 times. Irrigation water was supplied via a drip irrigation network using drippers (4.0 L h^−1^). All treated and untreated plants were exposed to 1 level of drought (65% of soil available water). Before treatment application, water available in sandy soil was calculated by determining the difference between field capacity (FC) and permanent wilting point (PWP) considering soil depth under dripping irrigation when zones of water overlap each other on the line. Time domain reflectometry (TDR) was used to determine available water and irrigation time. Drought was applied after the emerging fourth leaf of the plant and during the growth period. Plants were sprayed with a solution containing zinc and silicon nanoparticles at concentrations of 20 and 15 ppm, respectively, 3 times at an interval of 15 days. Zeolite nanoparticles were added at a rate of 1.3 L. ha^−1^ through the irrigation network 15 days after planting followed by foliar application after 15 days from the first application and 20 days from the second addition. Bentonite, kaolinite, and perlite were added to the soil 15 days before seed sowing at a rate of 12.5, 5, and 1.25 t.ha^−1^, respectively. The plant samples were selected randomly from each treatment and taken in different growth stages (vegetative growth, flowering stage, and seed production stage) for performing morpho-physiological and biochemical analysis as well as calculating the seed and oil yields.

### 4.3. Synthesis of Zeolite, Zinc, and Silicon Nanoparticles

#### 4.3.1. Zeolite Nanoparticles

Zeolite nanoparticles were prepared according to Li et al. [66] then were loaded with nitrogen (Figure 7A and Table 8) according to [67]. Transmission electronic microscope examination and imaging (TEM) were undertaken at the Research Park of the Faculty of Agriculture, Cairo University (FA-CURP).

#### 4.3.2. Zinc and Silicon Nanoparticles

All the used reagents were of analytical grade and the nanoparticles were prepared from their precursors. Zinc in the form of zinc chloride (ZnCl_2_) and Silicon in the form of Silicon tetrachloride (SiCl_4_) were purchased from Sigma Chemical Co. (St. Louis, MI, USA). Nanoparticles were obtained using the top-to-bottom molecular chemical method. Zinc ***nanoparticles*** (Figure 7B) were prepared from an aqueous solution of zinc chloride. Sodium hydroxide solution was added slowly in a molar ratio of 1:2 under vigorous stirring for 8 h. The obtained precipitate was filtered and washed thoroughly with ionized water in a mixed water/toluene system using a high-speed stirrer and then washed again with ionized water alone for 3 h. The precipitate was dried in an oven at 100 °C then exposed to 1.5 psi of pressure for 3 days discontinuously (7 h per day) [68].

As for silicon nanoparticles (Figure 7C), mild reagents ((3-aminopropyl) tri-methoxysilane and ascorbate sodium) were used through a quick reaction in a commonly used round bottom flask at room temperature and pressure. Trimethoxysilane (97%) and ascorbate sodium were prepared (1:4 m ratio) in an aqueous solution while stirring. Then, 1.25 mL of 0.1 M ascorbate sodium was added to the above mixture by stirring for 40 min, then exposed to 1.5 psi of pressure for 5 days discontinuously (8 h per day). The precipitate was dried in an oven at 80 °C for 10 h. [69].

### 4.4. Data Recorded

#### 4.4.1. Morphological Characteristics and Yield Components

After 15 days from treatments application, 10 drought-stressed coriander plants from each treatment were chosen randomly to determine the plant height, number of umbels, fresh weight dry weight, leaf area, and root length in both seasons.

#### 4.4.2. Leaf Chlorophyll Content, Photosynthetic Apparatus, and Chlorophyll Fluorescence Parameters

##### Leaf Chlorophyll Content and Photosynthetic Apparatus

Total chlorophyll content was measured by spectrophotometer and calculated according to the equation described by Moran et al. [70]. Measurements of net photosynthesis on an area basis [μmol CO_2_ m^−2^s^−1^], leaf stomatal conductance [mol H_2_O m^−2^s^−1^], intercellular CO_2_ concentration (ppm), and water use efficiency of 5 different leaves per treatment were monitored using a LICOR 6400 (Lincoln, NE, USA) infrared gas analyzer (IRGA). Light intensity (Photosynthetically active radiation, PAR) within the sampling chamber was set at 1500 [μmol m^−2^s^−1^], using a Li-6400- 02B LED light source (LI-COR). The CO_2_ flow into the chamber was maintained at a concentration of 400 μmol mol^−1^ using a LI-6400-01 CO_2_ mixer (LI-COR).

##### Measurement of Chlorophyll Fluorescence Parameters

A portable pulse amplitude-modulated chlorophyll fluorometer (FMS-2; Hansatech, King’s Lynn, UK) was used to measure chlorophyll fluorescence parameters, such as minimum chlorophyll fluorescence yield of the dark-adapted state(F0), maximal fluorescence yield of the dark-adapted state (FM), steady-state fluorescence yield (FS), minimum fluorescence of the light-adapted state (F0’), and maximal fluorescence yield of the light-adapted state (FM’). All measurements were taken 3 times. Under 800 mmol m^–2^ s^–1^ light, the leaves of each treated plant reached steady state after photochemistry and FS were measured; then, under saturated pulsed light (12,000 mmol m^–2^ s^–1^), FM’ was measured. Then the action light was closed and the far-red light was turned on immediately. F0’ was measured after 2 s. After that, dark treatment was carried out for 30 min with a dark adaptation clip. F0 and FM were measured.

#### 4.4.3. Total Carbohydrate Content

Total carbohydrates in plant herbs were determined using the phosphomolybdic acid method according to [71]. Approximately 2 gm of sample was crushed with 10 mL 80% ethanol in a mortar and pestle then filtered through Whatman filter paper. The filter and residue were collected separately. The alcohol residue was put into a 250 mL conical flask. A total of 150 mL distilled water and 5 mL concentrated HCl were added into the flask. The residue was hydrolyzed for 30 min and cooled to room temperature. Na_2_CO_3_ was then added slowly until the extract became neutral (pH = 7). The extract was filtered, and the residue was discarded. The filtrate was taken into a conical flask and condensed in a water bath for 3–4 min. Distilled water was added to the filtrate and was then filtered after the mixing residue was discarded and the volume of filtrate was served for reducing sugar. A total of 20 mL of this filtrate was put into a 150 mL conical flask and 2 mL of concentrated HCl was added to it. It was then hydrolyzed for 30 min and cooled at room temperature. Na_2_CO_3_ was slowly added until the extract became neutral (pH = 7). This extract was filtered, residue was discarded, and the final volume of the filtrate was measured and served as a sample for total sugar. Then, 0.5 mL of aliquot sample was taken into a test tube and 1 mL of Somogy’s reagent was added. Test tubes were placed in a boiling water bath for 30 min, cooled to room temperature, and 1 mL of arsenomolybdate reagent was added. The content was mixed and diluted to a volume of 10 mL and its absorbance was spectrophotometerically measured at 560 nm. Concentrations of Fe, Zn, Mn, and B in plant samples were determined using an atomic absorption spectrophotometer with air-acetylene and fuel (PyeUnicam, model SP-1900, International Equipment Trading Ltd., Mundelein, IL, USA).

#### 4.4.4. Phytohormones, Malondialdehyde, and Catalase

Freeze-dried plant herbs (6 g FW) were ground to a fine powder within a mortar and pestle. The powder was extracted 3 times (once for 3 h and twice for 1 h) with methanol (80% *v*/*v*, 15 mL/g FW) supplemented with butylated hydroxytoluene DBPC (2, 6-di-tert-butyl-P-crosol) as an antioxidant at 4 °C in darkness. The extract was centrifuged at 4000 rpm. The supernatant was transferred into flasks wrapped with aluminum foil and the residue was extracted twice. The supernatants were gathered and the total volume was reduced to 10 mL at 35 °C under vacuum. The aqueous extract was adjusted to pH 8.6 and extracted 3 times with an equal volume of pure ethyl acetate. The combined alkaline ethyl acetate extract was dehydrated over anhydrous sodium sulfate then filtered. The filtrate was evaporated to dryness under vacuum at 35 °C and redissolved in 1 mL absolute methanol. The methanol extract was used after methylation according to Fales et al. [72] to determine gibberellic acid (GA), abscisic acid (ABA), and indole-acetic acid (IAA). The quantification of the endogenous phytohormones was carried out with an *ATI*Unicam Gas Liquid Chromatography, 610 Series, equipped with a flame ionization detector according to the method described by [73]. The fractionation of phytohormones was conducted using a coiled glass column (1.5 m × 4 mm) packed with 1% OV-17. Gases flow rates were 30, 30, and 330 mL/min for nitrogen, hydrogen, and air, respectively. The peak identification and quantification of phytohormones were performed by using external authentic hormones and a Microsoft program to calculate the concentrations of the identified peaks.

For determining malondialdehyde (MDA) level, approximately 50 mg of freeze-dried samples were powdered in 0.5 mL of 0.1% (*w*/*v*) trichloroacetic acid. The resulting mixture was filtered and incubated with 0.5% (*v*/*v*) thiobarbituric acid (TBA) in 20% trichloroacetic acid at 95 °C. The absorbance was measured spectrophotometrically at 440-, 532-, and 600-nm waves using a microplate reader (Infinite 200 PRO, Tecan Group Ltd., Männedorf, Switzerland) [74].

Catalase activity (CAT) was determined using a spectrophotometric method defined by [75]. This method is based on the measurement of the decline in absorbance upon hydrogen peroxide cleavage at 240 nm. The measured difference is in absorbance per minute.

#### 4.4.5. Antioxidant Compounds Content and Proline

Total phenolic contents of the extracts were determined spectrophotometrically according to the Folin–Ciocalteu colorimetric method [76]. Total flavonoids were determined using the colorimetric process stated by Meda et al. [77]. Thiamin and ascorbic acid were determined in fresh herbs and estimated per 100 mL fresh weight fluorometrically [78] for detailed protocols. Carotenoid content in the plant leaves was extracted and quantified based on the technique of Lichtenthaler and Buschmann [79], which is published elsewhere [80].

Free proline content was extracted from the leaf tissues according to the method described by [81]. A cold extraction procedure was used by mixing 20–50 mg fresh weight aliquots with 0.5–1 mL of ethanol: water (60:80 *v*/*v*). The resulting mixture was left overnight at 5° C and then centrifuged at 15,000× *g* (4 min). A total of 1 mL of alcoholic extract was diluted with 10 mL of distilled water. Then, 5 mL of ninhydrin (0.125 g ninhydrin, 2 mL of 6 mM NH_3_PO_4_, 3 mL of glacial acetic acid) and 5 mL of glacial acetic acid were added, and the mixture was placed in a boiling water bath for 45 min at 100 °C. The reaction was stopped by placing the test tubes in cold water. The cold extraction procedure was repeated on the pellet, and then the pooled supernatants were used for the analysis using a PD-303 model spectrophotometer.

#### 4.4.6. Plant Nutrient Content

The plant material was dried in an electric oven at 70 °C for 24 h according to [82], then finely ground for chemical determination of elements. The wet digestion of 0.2 g plant material with sulphuric and perchloric acids was carried out on herbs by adding concentrated sulfuric acid (5 mL) to the samples, and the mixture was heated for 10 min. Then, 0.5 mL perchloric acid was added and heating continued until a clear solution was obtained [64,71]. Total nitrogen content of the dried leaves was determined using the modified micro-Kjeldahl method as described by [64]. Phosphorus was determined colorimetrically by using the chlorostannous molybdophosphoric blue colour method in sulphuric acid according to [71]. Potassium, sodium, and calcium concentrations were determined using flame photometer apparatus (CORNING M 410, Sherwood Scientific Ltd., Cambridge, UK).

#### 4.4.7. Essential Oil Content

Ripened fruits were collected from each treatment and were air dried and weighed to extract the essential oil in both seasons. Dry fruits (300 g) from each replicate of all treatments were subjected to hydro-distillation for 3 h using a Clevenger-type apparatus [82]. The essential oil content was calculated as a percentage. Essential oil ml/plant and liter/feddan were calculated according to the dry weight of fruits.

#### 4.4.8. Gas Chromatography-Mass Spectrometry

The ADELSIGLCMS system (Agilent Technologies, Palo Alto, CA, USA), equipped with a BPX5 capillary column (0.22 mm id × 25 m, film thickness 0.25 μm) was used. Analysis was carried out using helium as the carrier gas, with the flow rate at 1.0 mL/min. The column temperature was programmed from 60 °C to 240 °C at 3 °C/min. The sample size was 2 μL, and the split ratio was 1:20. The injector temperature was 250 °C. The ionization voltage applied was 70 eV, and the mass range was *m/z* 41–400 amu. Kovat’s indices were determined by co-injection of the sample with a solution containing a homologous series of n-hydrocarbons in a temperature run identical to that described above. The separated components of the essential oil were identified by matching them with the National Institute of Standards and Technology (NIST) mass spectral library data, through comparison of the Kovat’s indices with those of authentic components, and comparison with published data [83]. The quantitative determination was carried out based on peak area integration.

### 4.5. Statistical Analysis

The data obtained in this study are the mean values of 4 replicates ± standard error (S.E). The significant differences between treatments were tested via analysis of variance (one-way ANOVA) using Statistica 7 software. To compare means, Tukey’s tests at *p* ≤ 0.05 were implemented. Pearson’s correlation analysis and a heatmap were completed by an online platform for data analysis and visualization [84].

## 5. Conclusions

The present study proved that drought stress affects growth characteristics, physiological properties, and biochemical reactions of coriander plants. Nevertheless, the application of zeolite, silicon, and zinc nanoparticles has revealed their ability to reduce the harmful effects of water deficit through the improvement of photosynthetic pigments, photosynthesis rate, water use efficiency, photosystem II efficiency, chlorophyll fluorescence, stomatal conductance, modulating plant hormones (IAA, GA3, and ABA), and ameliorating nutrient uptake, proline content, and non-enzymatic antioxidant systems. On the contrary, these treatments significantly reduced lipid peroxidation (MDA) by 16.2, 6.19, and 8.25% in the first season and 37.32, 4.71, and 9.96% in the second season, respectively. According to the data of this study, the application of zeolite, silicon, and zinc nanoparticles could be a promising solution for enhancing plant growth and seed and oil productivity in arid and semi-arid areas and become a vital means for agriculture production system sustainability in facing climate change.

## Figures and Tables

**Figure 1 plants-12-02838-f001:**
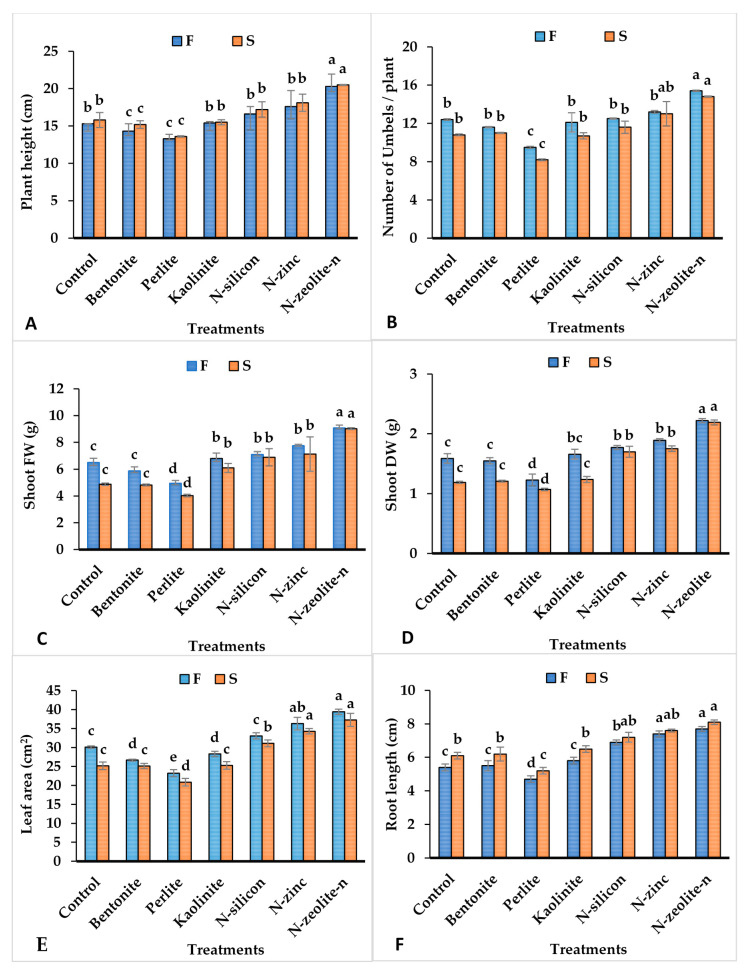
Influence of different treatments on plant height (**A**), number of umbels (**B**), fresh weight (**C**), dry weight (**D**), leaf area (**E**), and root length (**F**) of coriander plants grown under drought stress conditions. Columns followed with the different letters point out different significance levels between treatments according to the Tukey HSD test (*p* ≤ 0.05). Bar above columns indicates standard deviation. F = first season, S = second season.

**Figure 2 plants-12-02838-f002:**
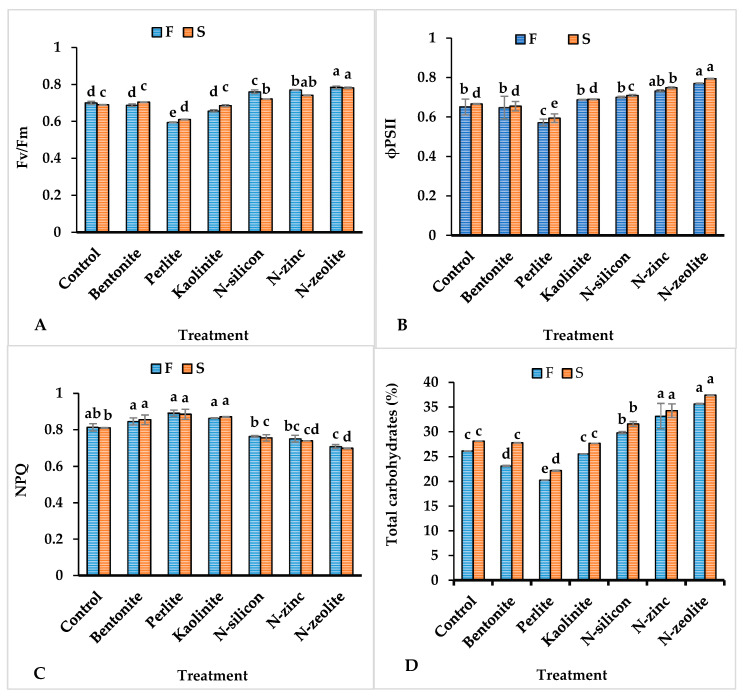
Influence of different treatments on Chlorophyll fluorescence (Fv/Fm—**A**), photosystem II efficiency (ϕPSII—**B**), nonphotochemical quenching (NPQ—**C**), and total carbohydrate content (**D**) of coriander plants grown under drought stress conditions. Columns followed with different letters point out different significance between treatments according to the Tukey HSD test (*p* ≤ 0.05). Bar above columns indicates standard deviation. F = first season, S = second season.

**Figure 3 plants-12-02838-f003:**
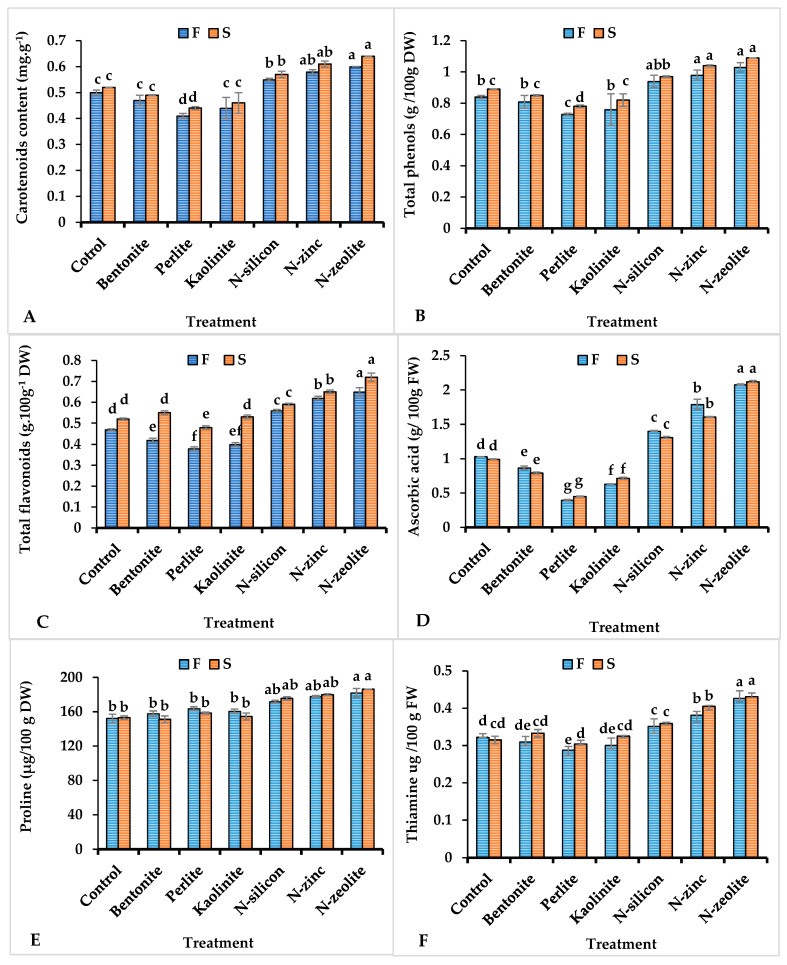
Influence of different treatments on leaf carotenoids (**A**), total phenols (**B**), total flavonoids (**C**), ascorbic acid (**D**), proline (**E**), and thiamine content (**F**) of coriander plants grown under drought stress conditions. Columns followed with different letters point out different significance between treatments according to the Tukey HSD test (*p* ≤ 0.05). Bar above columns indicates standard deviation. F = first season, S = second season.

**Figure 4 plants-12-02838-f004:**
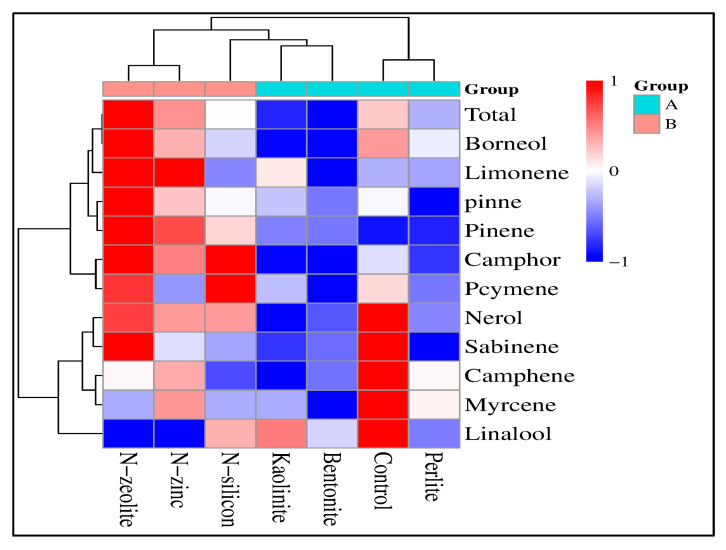
Heatmap correlation between fatty acid compounds of coriander plants grown under drought stress conditions and treated with bentonite, perlite, kaolinite, N-silicon, N-zinc, and N-zeolite.

**Figure 5 plants-12-02838-f005:**
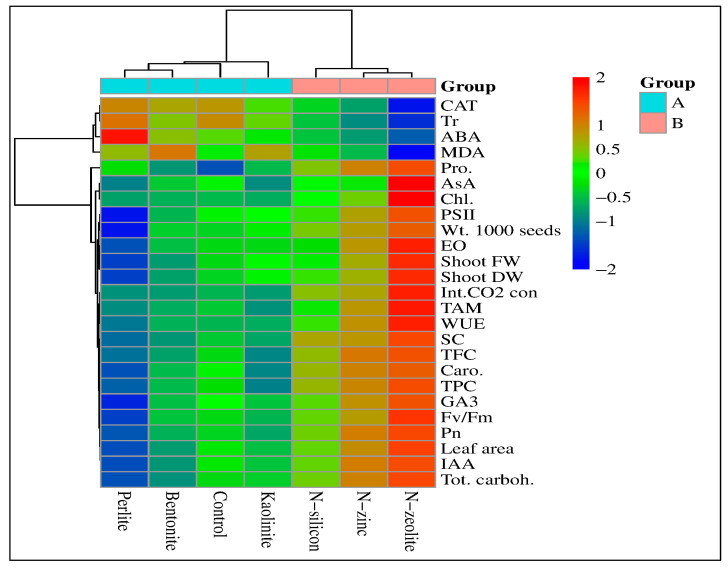
Heatmap correlation between morphological and physiochemical properties of coriander plants grown under drought stress conditions and treated with bentonite, perlite, kaolinite, N-silicon, N-zinc, and N-zeolite. Abbreviations: CAT, catalase; ABA, abscisic acid; MDA, malondialdehyde; Pro, proline; AsA, ascorbic acid; Caro, carotenoid content; IAA, indole-3-acetic acid; TAM, thiamine; LA, leaf area; Shoot FW, shoot fresh weight; Shoot DW, shoot dry weight; TPC, total phenol content; TFC, total flavonoid content; Chl, chlorophyll content; Pn, photosynthesis rate; Wt. 1000 seeds, weight of 1000 seeds; EO, essential oil; Int.CO_2_.con, intercellular CO_2_ concentration; WUE, water use efficiency; SC, stomatal conductance; Fv/Fm, chlorophyll fluorescence, photosystem II efficiency (PSII), and Tot.carboh., total carbohydrate content.

**Figure 6 plants-12-02838-f006:**
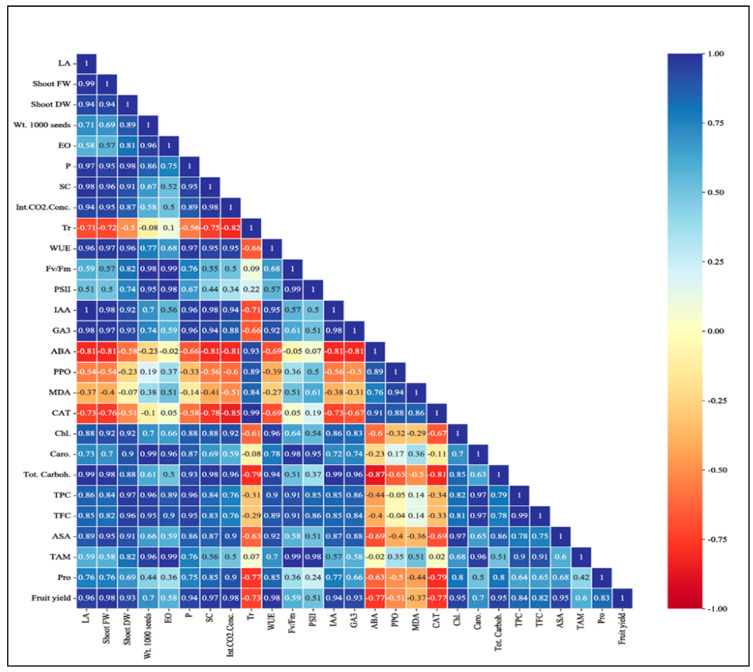
Pearson’s correlation analysis between morphological and physiochemical properties of coriander plants grown under drought stress conditions and treated with bentonite, perlite, kaolinite, N-silicon, N-zinc, and N-zeolite. Abbreviations: CAT, catalase; ABA, abscisic acid; MDA, malondialdehyde; Pro, proline; AsA, ascorbic acid; Caro, carotenoid content; IAA, indole-3-acetic acid; TAM, thiamine, LA, leaf area; Shoot FW, shoot fresh weight; Shoot DW, shoot dry weight; TPC, total phenol content; TFC, total flavonoid content; Chl, chlorophyll content; Wt. 1000 seeds, weight of 1000 seeds; EO, essential oil; Int.CO_2_.Conc, intercellular CO2 concentration; WUE, water use efficiency; SC, stomatal conductance; Fv/Fm, chlorophyll fluorescence, photosystem II efficiency (PSII); Pro, proline; and Tot.Carboh., total carbohydrate content.

**Figure 7 plants-12-02838-f007:**
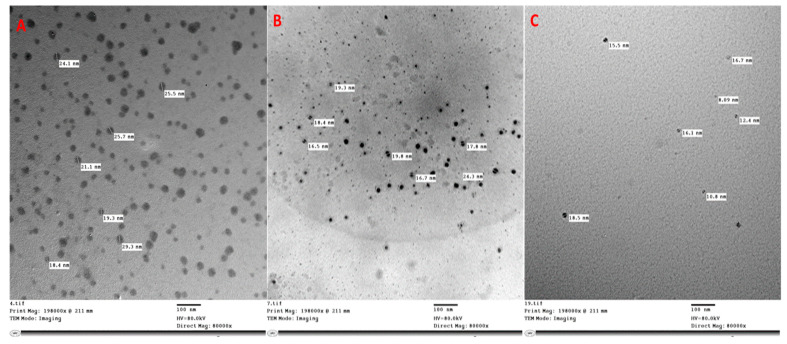
Scanning electron microscopy (SEM) for synthesized zeolite (**A**), zinc (**B**), and silicon (**C**) nanoparticles.

**Table 1 plants-12-02838-t001:** Influence of different treatments on photosynthetic machinery of coriander plants grown under drought stress conditions.

Treatment	Chlorophyll Content (mg.g^−1^)		Photosynthetic Rate (µmol m^–2^s^–1^)		Stomatal Conductance (mmol m^–2^s^–1^)		Intercellular CO_2_ Concentration (ppm)		Transpiration Rate (mmol m^–2^s^–1^)		Water Use Efficiency (μmol mmol^−1^)	
	F	S	F	S	F	S	F	S	F	S	F	S
Control	2.28 c	2.36 c	5.309 b	4.751 d	33.24 c	31.22 c	138.91 c	125.18 c	7.721 a	8.053 a	0.687 c	0.589 c
Bentonite	2.26 c	2.39 c	4.662 c	4.110 d	30.18 c	32.19 c	131.45 c	122.63 c	7.11 b	7.690 b	0.655 c	0.534 c
Perlite	2.02 d	2.18 d	3.204 c	3.109 d	27.61 c	30.28 c	126.71 c	115.97 d	7.948 a	8.06 a	0.403 d	0.393 d
Kaolinite	2.24 c	2.31 c	4.530 c	4.085 d	31.15 c	30.92 c	130.44 c	123.09 c	6.981 a	7.215 b	0.648 c	0.566 c
N-silicon	2.45 b	2.49 b	6.859 b	6.937 c	42.11 b	43.04 b	188.29 b	191.04 b	5.886 b	5.210 c	1.165 b	1.331 b
N-zinc	2.57 b	2.69 b	8.207 a	7.914 b	43.21 b	42.89 b	196.88 b	206.37 a	5.282 b	5.079 c	1.553 b	1.558 b
N-zeolite	3.09 a	3.11 a	9.067 a	8.669 a	48.05 a	47.25 a	241.21 a	252.06 a	4.371 c	3.975 d	2.074 a	2.180 a

Means followed by different letters point to significant differences between the treatments at according to Tukey’s HSD test (*p* ≤ 0.05). F = first season, S = second season.

**Table 2 plants-12-02838-t002:** Influence of different treatments on phytohormones, malondialdehyde, and catalase of coriander plants grown under drought stress conditions.

Treatment	IAA (µg/g F.W/Leaves)		GA3 (µg/g F.W/Leaves)		ABA (µg/g F.W/Leaves)		MDA (Units mg^−1^ Protein)		CAT (Units mg^−1^ Protein)	
	F	S	F	S	F	S	F	S	F	S
Control	18.901 c	18.720 c	36.90 c	38.115 c	16.429 b	18.109 a	4.261 a	4.078 a	14.722 a	15.051 a
Bentonite	17.953 d	17.105 d	37.805 c	36.332 c	16.796 b	17.441 a	4.815 a	4.922 a	14.582 a	15.307 a
Perlite	15.908 e	14.880 e	32.107 d	28.311 d	19.028 a	19.504 a	4.510 a	5.200 a	14.903 a	15.009 a
Kaolinite	19.401 c	18.776 c	38.103 c	35.109 c	15.630 b	16.408 b	4.616 a	4.190 a	13.926 a	14.622 a
N-silicon	23.404 b	21.759 b	41.694 b	40.810 b	15.142 b	15.026 b	4.069 ab	4.008 ab	13.051 b	12.691 b
N-zinc	25.013 a	22.105 b	44.503 a	42.775 b	14.582 b	14.972 b	3.875 b	3.290 b	12.640 b	12.033 b
N-zeolite	26.428 a	25.337 a	46.968 a	46.115 a	13.708 c	13.994 c	3.103 b	2.795 b	10.976 c	10.351 c

Means followed by different letters point to significant differences between the treatments according to Tukey HSD test (*p* ≤ 0.05). F = first season, S = second season.

**Table 3 plants-12-02838-t003:** Influence of different treatments on endogenous macro and micronutrient accumulation in leaf tissue of coriander plants grown under drought stress conditions.

Treatment	N (%)		P (%)		K (%)		Mg (%)		Ca (%)		Fe (ppm)		Zn (ppm)		Mn (ppm)	
	F	S	F	S	F	S	F	S	F	S	F	S	F	S	F	S
Control	2.57 c	2.51 c	0.27 bc	0.25 c	3.28 b	3.21 c	0.22 c	0.20 c	1.48 c	1.44 c	161.2 c	160.5 c	48.8 bc	48.2 bc	10.6 c	11.c
Bentonite	2.56 c	2.57 c	0.27 bc	0.26 c	3.25 bc	3.24 c	0.23 c	0.21 c	1.41 c	1.42 c	168.4 c	165.2 c	46.61 c	49.5 bc	11.3 b	11.1 c
Perlite	2.32 d	2.42 d	0.25 c	0.23 c	3.20 d	3.22 d	0.20 d	0.20 d	1.42 d	1.29 d	155.7 d	152.9 d	45.2 c	47.6 c	10.8 c	11.0 c
Kaolinite	2.45 c	2.55 c	0.25 c	0.24 c	3.28 b	3.26 c	0.23 c	0.20 c	1.47 c	1.40 c	167.5 c	166.3 c	47.3 c	46.8 c	10.2 c	10.6 c
N-silicon	2.85 b	2.80 b	0.30 b	0.32 b	3.33 b	3.29 bc	0.30 b	0.32 b	1.51 b	1.50 b	181.2 b	180.3 b	52.5 b	50.4 b	13.2 b	13.0 b
N-zinc	2.96 b	3.12 a	0.30 b	0.34 b	3.43 a	3.47 b	0.33 b	0.35 b	1.53 b	1.55 b	187.5 b	185.6 b	56.4 b	58.2 a	13.3 b	13.5 b
N-zeolite	3.11 a	3.20 a	0.34 a	0.37 a	3.51 a	3.59 a	0.38 a	0.40 a	1.60 a	1.63 a	193.5 a	196.8 a	61.5 a	60.3 a	15.5 a	16.8 a

Means followed by different letters point to significant differences between the treatments according to Tukey’s HSD test (*p* ≤ 0.05). F = first season, S = second season.

**Table 4 plants-12-02838-t004:** Influence of different treatments on yield and its components in coriander plants grown under drought stress conditions.

Treatment	Fruit Yield/Plant (g)		Weight of 1000 Seeds (g)		Fruit Yield (Kg/fad)		Essential Oil/Plant (mL)		Essential Oil (L/fed.)	
	F	S	F	S	F	S	F	S	F	S
Control	12.16 b	12.00 b	8.37 c	8.01 c	302.41 c	300.39 c	0.050 c	0.052 c	3.10 c	3.09 c
Bentonite	12.15 b	12.57 b	8.35 c	8.02 c	300.79 c	298.55 c	0.051 c	0.049 c	3.05 c	3.03 c
Perlite	11.49 b	11.16 b	7.22 d	7.37 d	289.63 d	280.46 cd	0.042 d	0.040 d	2.78 d	2.72 d
Kaolinite	12.46 b	12.44 b	8.19 c	8.05 c	306.27 c	302.62 c	0.051 c	0.052 bc	3.24 c	3.13 c
N-silicon	13.25 ab	13.22 ab	8.89 b	8.86 b	328.13 b	325.10 b	0.058 b	0.059 b	3.26 b	3.19 b
N-zinc	13.32 ab	13.29 ab	9.12 a	9.10 a	337.45 b	335.72 b	0.060 ab	0.060 ab	3.75 ab	3.70 ab
N-zeolite	14.68 a	14.53 a	9.44 a	9.39 a	350.25 a	346.59 a	0.066 a	0.064 a	4.15 a	4.12 a

Means with the same letters in a column are not significantly different according to Tukey’s HSD test (*p* ≤ 0.05), F = first season, S= second season.

**Table 5 plants-12-02838-t005:** Influence of different treatments on seed essential oil composition of coriander plants grown under drought stress conditions.

Fatty Acids Compounds	Control		Bentonite		Perlite		Kaolinite		N-Silicon		N-Zinc		N-Zeolite	
	F	S	F	S	F	S	F	S	F	S	F	S	F	S
β-Pinene	2.46	2.51	3.11	2.96	2.31	2.79	3.07	3.15	4.05	4.12	4.71	5.03	6.19	6.88
Limonene	3.66	3.97	2.71	2.89	3.45	4.11	4.03	4.29	3.59	3.78	5.02	4.94	5.15	5.44
P-cymene	4.11	4.75	3.15	3.55	3.82	4.09	4.12	4.18	5.44	5.48	3.89	4.18	4.62	5.1
α-pinne	3.29	3.41	3.02	3.13	2.08	3.16	3.21	3.27	3.32	3.37	3.46	3.52	4.15	4.63
Camphor	2.78	2.81	2.06	2.39	2.36	2.48	2.26	2.38	3.54	3.88	3.13	3.2	3.26	3.71
Nerol	0.72	0.81	0.45	0.51	0.42	0.44	0.55	0.61	0.52	0.58	0.33	0.39	0.35	0.37
Camphene	0.35	0.36	0.25	0.21	0.22	0.26	0.3	0.32	0.28	0.3	0.27	0.31	0.29	0.33
Sabinene	0.55	0.57	0.31	0.46	0.4	0.45	0.32	0.37	0.36	0.4	0.41	0.48	0.42	0.44
Myrcene	0.5	0.54	0.39	0.44	0.37	0.42	0.38	0.43	0.4	0.46	0.41	0.47	0.49	0.51
Borneol	0.15	0.17	0.05	0.08	0.1	0.12	0.09	0.11	0.07	0.12	0.1	0.13	0.09	0.1
Linalool	44.17	46.22	28.74	28.96	40.26	41.36	31.36	33.48	39.24	40.52	44.80	45.16	57.39	58.26
Total%	62.74	66.12	44.24	45.58	55.79	59.68	49.69	52.59	60.81	63.01	66.53	67.81	82.4	85.77

**Table 6 plants-12-02838-t006:** Physicochemical properties of experimental soil.

Parameters	Soil Depth (cm)	
	0–30	30–60
**Particle size distribution (%)**		
Sand	90.10	90.00
Silt	6.90	6.50
Clay	3.00	3.50
Textural class	Sand	Sand
Saturation water content (cm^3^ cm^−3^)	0.385	0.396
Field capacity cm^3^ cm^−3^	0.213	0.218
Permanent wilting point (cm^3^ cm^−3^)	0.057	0.057
Available water (cm^3^ cm^−3^)	0.156	0.161
Bulk density (mg m^−3^)	1.64	1.65
Saturated hydraulic conductivity, cm day^−1^	240.00	234.00
Organic matter (%)	0.31	0.25
Calcium carbonates (%)	4.80	3.71
pH (1:1, soil: water suspension)	7.70	7.81
EC(1:1, soil: water extract) (dS.m^−1^)	1.62	1.83
**Soluble Cations Cmole(+). Kg^−1^ soil**		
Ca^2^+	13.85	13.41
Mg^2+^	12.15	10.59
Na^+^	8.10	10.25
K^+^	6.00	6.05
**Soluble Anions, Cmole(−). Kg^−1^ soil**		
CO_3_^−^	-	-
HCO_3_^−^	11.92	9.75
Cl^−^	14.00	10.50
SO_4_^−^	15.08	21.30
**Available nutrients mg Kg^−1^ soil**		
N	16.21	13.12
P	7.78	6.21
K	46.50	45.89
Fe	9.20	12.00
Mn	1.63	1.50
Cu	2.10	1.15
Zn	2.00	1.61
B	0.23	0.21

**Table 7 plants-12-02838-t007:** Chemical properties of used compost.

Property	Value
Moisture content (%)	25
PH (1:5)	7.5
EC (1:5 extract) dsm^−1^	3.1
Organic-C (%)	33.11
Organic matter (%)	70
Total-N (%)	1.82
Total-K (%)	1.25
C/N ratio	14:1
Total-P (%)	1.29
Fe (ppm)	1019
Mn (ppm)	111
Cu (ppm)	180
Zn (ppm)	280
Total content of bacteria (cfu.g^−1^)	2.5 × 10^7^
Phosphate dissolving bacteria (cfu.g^−1^)	2.5 × 10^6^
Weed seeds	0

**Table 8 plants-12-02838-t008:** Chemical composition of zeolite nanoparticles loaded with nitrogen.

**Chemical** **Composition (%)**	**SiO_2_**	**TiO_2_**	**Al_2_O_3_**	**Fe_2_O_3_**	**FeO**	**MnO**	**MgO**	**CaO**	**Na_2_O**	**K_2_O**	**SrO**	**P_2_O_3_**	**N**
	45.50	2.81	13.30	5.40	8.31	0.51	6.30	9.52	2.83	0.87	0.22	0.67	2.70
**Trace elements** **(ppm)**	**Ba**	**Co**	**Cr**	**Se**	**Cu**	**Zn**	**Zr**	**Nb**	**Ni**	**Rb**	**Y**		
	10	1.2	35	0.8	19	64	257	13	55	15	22		

## Data Availability

Not applicable.

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
