# Peer review of "Application of Silicon, Zinc, and Zeolite Nanoparticles—A Tool to Enhance Drought Stress Tolerance in Coriander Plants for Better Growth Performance and Productivity"

_plants, 2023, doi:10.3390/plants12152838_

Round 1
Reviewer 1 Report
The nanotechnology is steadily transforming the modern agriculture. One of the promising strategies is the use of nanoscale agrochemicals, such as nanofertilizers and nanopesticides which have already yielded gratifying results in reducing fertilizers and pesticides, and also in precise agricultural practice. In this manuscript, the authors demonstrated that the application of silicon, zinc, and zeolite nanoparticles positively influenced the morphological, physiological, and biochemical properties of the drought-stressed coriander plant, which could be a promising approach to improve plant growth and productivity as well as to alleviate the adverse impacts of drought stress on coriander plants in arid and semi-arid areas. The findings and manipulations in this study may be beneficial to the readers of this Journal. However, there are still much room for improvement in the writing, standard expression and technical terminology.
Major issues
1. The definition and terminology of the nanoparticles used in this study are of confusion and inconsistency. e.g., The terms “zeolite nanoparticle”, “zeolite nanoparticulate”, “nano-zeolite”, “nano-zeolite-n” and “N-zeolite-n” are all used in this manuscript, but the authors did not define them and did not reveal their relationships.
2. There are certain imperfections in the experimental design. Since the purpose of this study is to assess the effects of Silicon, Zinc and Zeolite Nanoparticulates, the Silicon, Zinc and Zeolite in conventional form should be used in the experimental design, ensuring the equivalent supply of water and nutrients in the experimental pairs.
3. Did “N-zeolite-n” treatment supply more nitrogen to the plant than that of other treatment? Is the extra nitrogen main reason for its best results on biological index?
4. In line 15, “nan-element” and in line 44, “Nano-element”, which one is correct? Please define “Nano-element”. It did not appear in the main text.
Minor issues
1. In some paragraphs, punctuation is misused, resulting ambiguity in some sentences. For example:
Line 118, “N-silicon N-zinc”. No comma between the two words? Please also check Line 171, 358, 364, 371, 410, 426, 434, and 532.
Line 120, “stress conditions. studied growth”.
Line 552, “coriander plants [61-63]”. No stop at the end.
Line 606 “Synthesis of Zeolite Zinc and Silicon Nanoparticles”. No comma?
2. Line 633, “synthesized (A) zeolite, zinc, and (B) silicon (C) nanoparticles”. Please check the position of (A), (B), and (C).
3. Line 666, “HCL” should be “HCl”.
Author Response
Please open the attached file.

Reviewer 2 Report
The manuscript Application of Silicon, Zinc and Zeolite Nanoparticulate - A Tool to Enhance Drought Stress Tolerance in Coriander Plants for Better Growth Performance and Productivity’’ is looking to overcome the challenges of plant growth in low-water conditions using nanoparticles! The idea of the study is good, and it is an interesting topic somehow. However, it is not unique and there are some points that need to be improved.
Abstract
The abstract is tooooo long! Please see the instruction for authors on the MDPI website. It should be with Max. of 200 words!
Line 19: nano-zeolite (1.3 L.ha-1), nano-silicon (2.5 L.ha-1), and nano-zinc (2.5 L.ha-1). -> Why they are in L ha-1??? Should be in t ha-1?
Introduction
Line 76: [7-17]. -> Please avoid mentioning too many references at the end of a single paragraph! You mentioned 10 references here just in a pair of brackets! Make the text more specified and distribute the references into the paragraph.
So do like this: ‘In this regard, several agronomic strategies have been suggested to improve WUE and yield production of water stressed plants, for example, application of nanofertilizers [], utilization of microorganisms [], soil addition of organic fertilizers [], and biochar [], use of tolerant rootstocks and cultivars [], and implementation of material with high water retention, anti-transparent substances, enhancement of soil management, and application of efficient irrigation methods, in particular, in soils characterized by low water hold capacity [].
Line107-110: Same as the last comment and one thing more; Please avoid using unnecessary references and use lesser but related ones! This reference fits with your context you can add it here: https://doi.org/10.1007/s11270-022-05910-4
Results
Please check the format style for subsections and other parts of the manuscript based on MDPI instructions!
The quality of the figures is very poor. Please use the high-resolution version of them!
Discussion
Lines 417-425: this section is like an introduction! You talked already about drought stress in the introduction section and again here you are repeating that information! Why? In the discussion section go straight on interpreting your findings. That would be fine!
Lines 425-427: same as the last comment! why you are repeating the aims of the study again here?
This paragraph should be removed totally!
Line 431: [7,8,10,11,12,15]. -> Distribute these references in front of related plants in this part!
Line 461: negative impacts of drought by enhancing hydraulic conductivity -> Here is a good reference that fits with the effect of zeolite on hydraulic conductivity you can add it here: https://doi.org/10.3390/w14213506.
Material & Methods are well explained.
Concussion
Please add some values in percentages in this section.
Some minor spell checks are needed!
Author Response
Please open the attached file.

Round 2
Reviewer 2 Report
The manuscript considerably changed and it is recommended to be published!